# Temperature-Dependent Optical Properties of Perovskite Quantum Dots with Mixed-A-Cations

**DOI:** 10.3390/mi13030457

**Published:** 2022-03-17

**Authors:** Lei Hu, Weiren Zhao, Weijia Duan, Guojie Chen, Bingfeng Fan, Xiaoli Zhang

**Affiliations:** 1Guangdong Provincial Key Laboratory of Information Photonics Technology, Guangdong University of Technology, Guangzhou 510006, China; huleisichuandazhou@126.com (L.H.); dwj2862958509@126.com (W.D.); 2Guangdong-Hongkong-Macao Joint Laboratory for Intelligent Micro-Nano Optoelectronic Technology, School of Physics and Optoelectronic Engineering, Foshan University, Foshan 528225, China; gjchen@fosu.edu.cn

**Keywords:** perovskite quantum dots, mixed-cation doping, low-temperature PL

## Abstract

In this work, metal halide perovskite quantum dots (QDs) with Formamidinium (FA) and Cs mixed cations were fabricated using a solution-processed method at room temperature. By controlling Cs doping ratios in a precursor, the optical properties of mixed-cation perovskite QDs were systematically studied. With the increase in Cs ion doping, the photoluminescence (PL) spectra of perovskite QDs were blueshifted, which was mainly due to the smaller radius of Cs ions than those of FA. Temperature-dependent PL spectra were conducted on mixed-cation perovskite QDs. As the temperature gradually increased from 4 K to 300 K, PL peaks were blue shifted, and full-width at half maximum (FWHM) was widened, which was directly related to lattice thermal expansion and the carrier-photon coupling effect under temperature variation. At the same time, excess Cs ion doping had a prominent influence on optical properties at low temperatures, which was mainly due to the introduction of detrimental defects in perovskite crystals. Therefore, it is particularly important to control doping concentration in the preparation of high-quality perovskite QDs and efficient photoelectric devices.

## 1. Introduction

Metal halide perovskite semiconductors have emerged as one of the most promising materials for application in optoelectronic devices over the past years. As a new generation of photoelectric material, perovskite is widely used in solar cells, light-emitting diodes, detectors, photocatalysis, lasers and other fields [1,2,3,4,5]. Perovskite QDs have excellent photoelectric properties, including tunable wavelength in the visible range, high quantum yield, a facile synthesis method, low cost, high absorptivity, a long charge diffusion length and high carrier mobility [6,7,8,9,10]. In order to improve the performance and stability of perovskite optoelectronic devices, researchers at home and abroad have proposed various research methods. For example, by adjusting the interface layers to reduce the defects, the performance of the device can also be improved while enhancing the carrier transport efficiency [11]; by optimizing the structure of optoelectronic devices, so that the energy level of the device is more consistent with that of the perovskite material, with further improvement to the efficiency of optoelectronic devices by reducing the potential barrier of carrier transportation [12,13]; and by preparing all inorganic perovskite optoelectronic devices, which can alleviate the water–oxygen instability of perovskite materials to a certain extent, so as to improve the overall performance of the devices [14,15].

Metal halide perovskite has the general formula of AMX_3_, in which A cations are generally CS, FA (CH_3_(NH_3_)_2_) and MA (CH_3_NH_3_); M cations are metal cations, such as Pb, Sn, etc.; and X sites are halogen ions, including Cl, Br and I. By adjusting the halogen ion at the X position, the wavelength of perovskite QDs can be adjusted from blue to red, covering the whole visible range [16,17,18]. In addition, the size of perovskite QDs can be adjusted by synthesizing temperature, growth time and ligand adjustment, thus controlling the luminous wavelength [19,20,21]. In order to improve the efficiency and stability of perovskite optoelectronic devices, smaller Cs (or MA) ions were doped into perovskite with A ions of large radii. For example, when smaller Cs cations were doped into the perovskite QDs FAPbBr_3_, the luminescence of LED could realize cd m^−^^2^ at the optimized doping concentration [22]. Doping a small amount of MA ions into the perovskite of FAPbI_3_ can improve the crystallinity of the perovskite film, so that the photoelectric conversion efficiency of solar cells can exceed 20% [23]. It has been reported that the photoelectric conversion efficiency of two-cation- or even tri-cation-doped perovskite solar cells is significantly improved [24,25].

Cation doping has great effects on the excellent optoelectronic properties of perovskite devices. In view of this, this paper used a method to synthesize perovskite QDs with mixed cations, and conducted an in depth analysis. To study the effects of cation doping on the optical properties, temperature-dependent PL was analyzed. Through this work, we can systematically understand the influence of cations doped on the optical properties of FA-based perovskite QDs, providing support for the preparation and analysis of efficient optoelectronic devices.

## 2. Materials and Methods

### 2.1. Materials

CsBr (99%); FABr (99%); PbBr_2_ (99%); chemical solvents DMF (Dimethy Formamide) and DMSO (Dimethyl Sulfoxide); toluene; oleic acid (OA); and oleylamine (OAm) were purchased from Aladdin, Shanghai, China and Shenzhen Huatest Technology Co., Ltd., Shenzhen, China, respectively. In the preparation of perovskite QDs, all raw materials were directly used, without any treatment and purification.

### 2.2. Synthesis

Figure 1 shows the synthesis process of perovskite QDs. First, 5 mmol FABr and 5 mmol PbBr_2_ were dissolved in a 5 mL mixture of DMF and DMSO (mixing ratio of 7:3), under the action of magnetic stirring, to obtain a clear precursor solution. Then, the precursor was added into a beaker containing 10 mL toluene solvent, drop by drop. At the same time, an appropriate amount of OA and OAm were dropped into the mixture. The color of the mixture instantly turned from transparent to green. Finally, the green solution was centrifuged at 5000 rpm for 5 min, followed by collection of the obtained supernatant in the reagent bottle for subsequent testing and analysis. For Cs ion-doped perovskite QDs, CsBr, FABr and PbBr_2_ were proportionally added to the mixture of DMF and DMSO. The selected ratios were FA:Cs of 0.90:0.10, 0.85:0.15 and 0.75:0.25, respectively. The concentration of perovskite QDs in our work was about 20 mg mL^−^^1^, which was quantified by drying 100 μL products. The synthesis process was consistent with that of FAPbBr_3_ QDs, and all experimental parameters were guaranteed to be consistent. We dipped a cotton swab into the perovskite QDs solution and wrote the word “GUST” on the slide, forming a thin film. The word, excited by an ultraviolet flashlight, emitted a green light, as shown in the photo.

### 2.3. Measurement

The as-prepared perovskite QDs were spin-coated on the glass or Si substrate, forming perovskite on this film for further characterization. The fluorescence characteristics of perovskite QDs were measured by fluorescence spectrometer (PL, C11347-12 (HAMAMATSU, Shenzhen, China)), and the morphology of the synthesized perovskite QDs was characterized by transmission electron microscopy (TEM, JEOL 2010, Shenzhen, China). The UV excitation wavelength was 365 nm from ultraviolet flashlight. The low-temperature PL was performed by an optical cryogenic thermostatic system (JANIS SHI-4, Shenzhen, China).

## 3. Results and Discussion

### 3.1. Morphology and Structure of Perovskite QDs

Through a simple solution-synthesis method, we obtained perovskite QDs at room temperature. In order to intuitively detect the morphology of perovskite QDs, we conducted a TEM test, as shown in Figure 2. From the figure, we can clearly see that the perovskite QDs are cubic in shape and uniformly dispersed, and the side length of the QD is about 10 nm. In general, perovskites have an orthorhombic phase, with the A-site cation at the center of the octahedron, while the B-site metal ions form an octahedral coordination with six halogen ions, as shown in schematic crystal-structure in Figure 2b. The structure of as-prepared perovskite QDs was detected by XRD patterns, as shown in Figure 2c, which indicates that the diffraction patterns of perovskite QDs are well matched with the cubic perovskite crystal. Using HRTEM images, we found that perovskite QDs have very good crystallinity. The TEM images and XRD patterns of perovskite QDs with Cs doping, are shown to be in Appendix A. By comparing different doping ratios of Cs ions, it is found that ion doping has some influence on the structure of perovskite QDs, indicating that Cs ion doping may introduce defects to a certain degree, which would be verified by varied luminescence spectra.

### 3.2. PL Spectra of Different Perovskite QDs

In order to observe the optical properties of perovskite QDs with mixed-cations, we conducted PL tests on all the samples, as shown in Figure 3. As the Cs cation ratio is gradually increased from 0 to 0.1, the luminescence peak of perovskite QDs decreases from 530 to 529 nm, and the FWHM changes from 19.3 to 20 nm. With the gradual increase in Cs ion doping ratio from 0.1 to 0.15 and 0.20, the emission wavelength of perovskite QDs is gradually blueshifted to 527 nm and 525 nm, and the FWHM changes to 19.6 nm and 19.1 nm, respectively. Figure 3b clearly shows the change rules of the FWHM and PL peak, where the abscissa from 1 to 4 represents the sample with Cs ion doping concentration increasing gradually. Obviously, the increase in Cs ion doping concentration will cause a PL blueshift in perovskite QDs (though the change is not very large) and also a change in FWHM. The main reason for this is that the radius size of a Cs ion (1.8 Å) is smaller than that of an FA ion (2.8 Å) [26]. According to Vegard’s law [27], Cs ion doping will lead to a smaller lattice constant, a change in the chemical bond between Pb and Br, and will further increase the bandgap width of perovskite, resulting in a PL blueshift. Therefore, Cs ion-doped perovskite QDs can regulate the luminescence wavelength of perovskite QDs within a small range, providing a new mean for bandgap engineering and wavelength regulation.

### 3.3. Temperature Dependent PL Spectra of Perovskite QDs

In order to study the optical properties of perovskite QDs and the influence of Cs ion-doping on perovskite QDs, we conducted low-temperature PL measurements on the samples, ranging from 4 K to 300 K, as shown in Figure 4 and Figure 5. When the temperature gradually rises from 4 K to 300 K, the luminescence peak of perovskite QDs gradually shifts to blue, corresponding to the increase in the bandgap width. This process involves a lattice thermal expansion effect and carrier-photon coupling effect [28]. As the temperature increases gradually, the lattice thermal expansion leads to a larger bandgap, which is manifested as a blueshift in the spectrum. Meanwhile, spectral broadening happens as the temperature is increased, which is related to carrier-photon coupling [28].

For Cs ion-doped perovskite QDs, the low-temperature spectrum is shown in Figure 5, in which Figure 5a–d indicate Cs ion-doping of 0, 0.1, 0.15 and 0.2, respectively. We found that as the temperature of the test system gradually increased from 4 K to 300 K, the PL spectral characteristics were consistent with that of law FAPbBr_3_ perovskite QDs; that is, the luminous peak was blueshifted, the bandgap was increased, and the FWHM was widened. This indicates that lattice thermal expansion and carrier-photon coupling effects are also applicable to perovskite QDs with mixed cations. However, by comparing the spectra, we can easily find that with the increase in Cs ion-doping concentration, a secondary PL peak appears. This indicates that Cs ion doping introduces defect energy levels into the lattice structure of perovskite, and with the increase in doping concentration, the influence of defect energy levels becomes prominent. At low temperatures, the effect of defects on luminescence is more obvious. As the temperature rises to 300 K, the luminescence of perovskite QDs is enhanced, and the influence of defects is gradually weakened. Therefore, proper selection of ion-doping concentration is crucial for the luminescence of perovskite QDs and the application of optoelectronic devices.

## 4. Conclusions

In this paper, perovskite QDs were prepared using a simple solution synthesis method at room temperature. By regulating Cs ion-doping in different proportions, we studied the effects of doping cation on the properties of perovskite QDs. The Cs ion radius is smaller than the FA ion radius, which causes the decreased lattice constant and increased bandgap, thereby resulting in the blueshift of the PL spectrum. As the temperature gradually rises from 4 K to 300 K, perovskite QDs undergo lattice thermal expansion and a carrier-photon coupling effect, resulting in PL blueshift and a broadening FWHM. Under the condition of low temperature, Cs doping-induced defects have a prominent influence on the optical properties. Therefore, it is very important to control and select the appropriate doping concentration for the preparation of efficient perovskite QDs and photoelectric devices.

## Figures and Tables

**Figure 1 micromachines-13-00457-f001:**
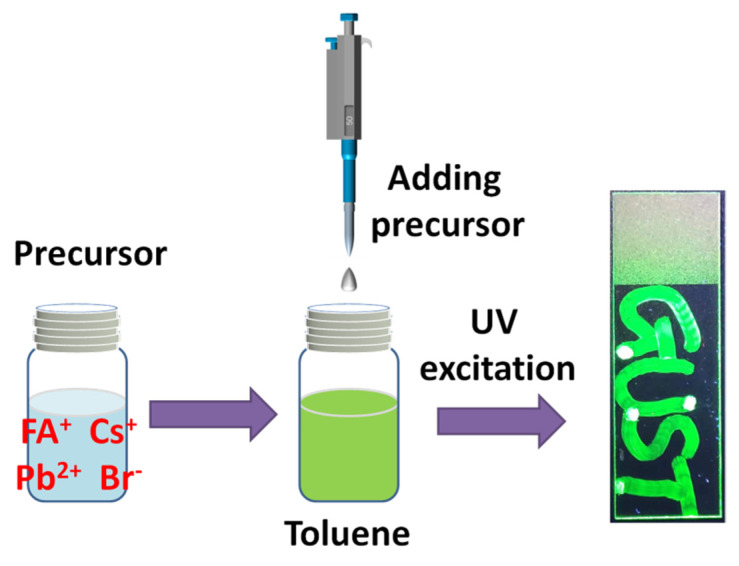
Schematic diagram of synthesis process of perovskite QDs.

**Figure 2 micromachines-13-00457-f002:**
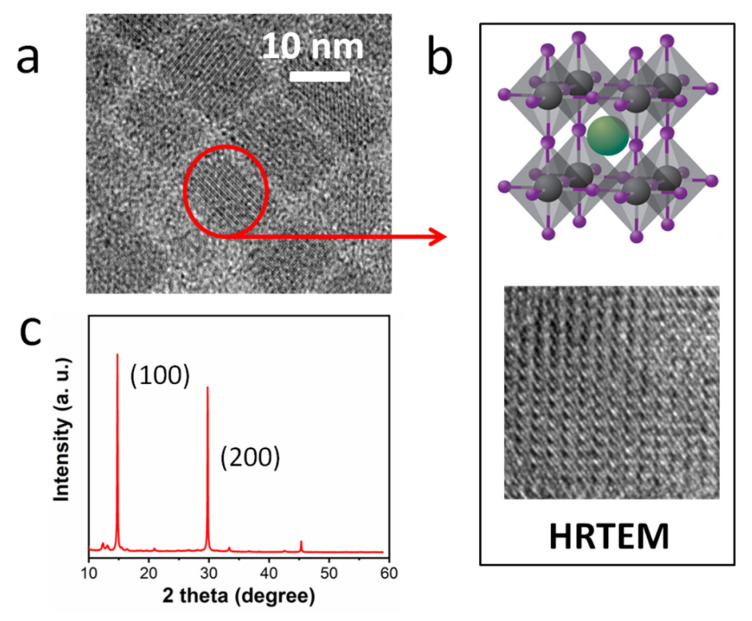
(**a**)TEM image, (**b**) structure diagram and HRTEM images, (**c**) XRD patterns of perovskite QDs (FAPbBr_3_).

**Figure 3 micromachines-13-00457-f003:**
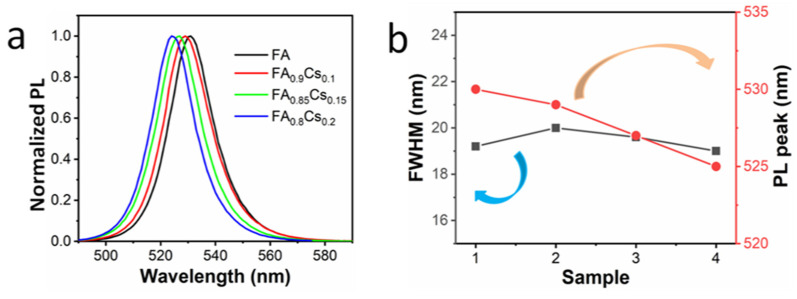
(**a**) PL spectra and (**b**) FWHM and PL peak changes in different perovskite QDs.

**Figure 4 micromachines-13-00457-f004:**
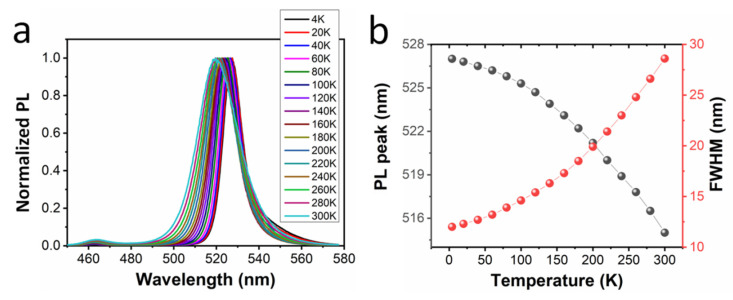
(**a**) Temperature-dependent PL spectra and (**b**) summarized PL peak and FWHM variations of perovskite QDs (FAPbBr_3_).

**Figure 5 micromachines-13-00457-f005:**
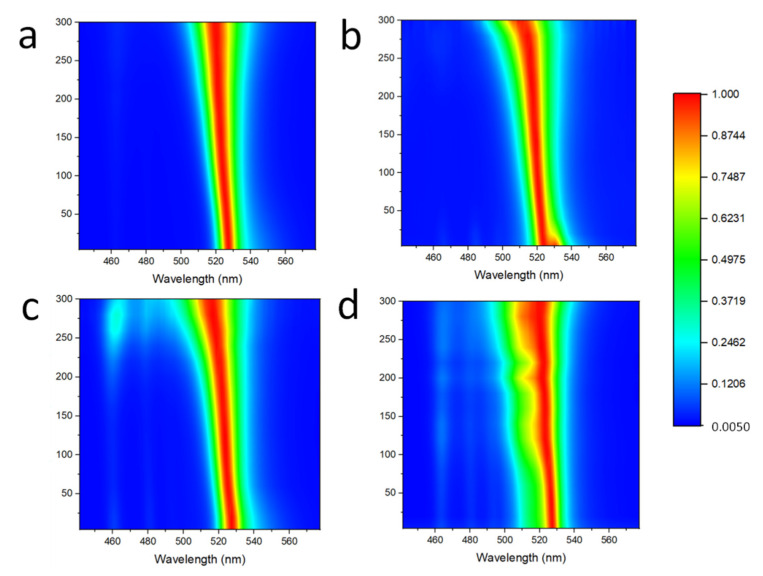
Temperature-dependent PL spectra of mixed-cation perovskite QDs: (**a**) Cs = 0; (**b**) Cs = 0.1; (**c**) Cs = 0.15; and (**d**) Cs = 0.2.

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
