# Peer review of "Temperature-Dependent Optical Properties of Perovskite Quantum Dots with Mixed-A-Cations"

_micromachines, 2022, doi:10.3390/mi13030457_

Round 1

Reviewer 1 Report

Please provide XRD and TEM results for all the samples used in Figure 4, to confirm that there is no impurity or morphology changes influences the temperature-dependent PL characterization results.

Reviewer 2 Report

In the manuscript "Temperature-dependent optical properties of perovskite quantum dots with mixed-A cations" the authors present a simple synthesis of alloyed CsMAPbBr3 nanocrystals and study their optical properties using temperature-dependent photoluminescence. The subject of study clearly lacks novelty as alloyed MA-Cs perovskite nanocrystals are well-known and studied. Also, the temperature-related PL is well described in the literature. In addition, there manuscript contains a series of errors, such as "dimethyl methylmaple", "oil amine", "orthonormal"system" and others. In the introduction the authors cite too many of their own articles in an unjustified manner. The XRD diffractogram presented on Fig. 2 is not described anywhere. Finally, the conclusions do not represent an important scientific insight or breakthrough. For these reasons, I do not recommend the manuscript for the publication.

Reviewer 3 Report

In this manuscript authors have synthesized both organic - inorganic and fully inorganic perovskite QDs and characterized them using temperature controlled PL and other techniques. Authors have done excellent work by synthesizing perovskite QDs at room temperature and using simple process. I have some supplementary questions regarding the process and sample preparation, which I believe will increase the value of this manuscript.

  1. Page 2 line 70: What is Oil Amine? Is it miss spelled?
  2. Can you please describe more clearly the synthesis process of perovskite QDs? Especially the final concentration of perovskite QDs.
  3. How did you prepare samples for characterization? Especially for PL, did you make thin film? 
  4. How to prepare QDs solution? How did you use the supernatant and where? 

These questions must be addressed before final publication.

Round 2

Reviewer 3 Report

Thanks to the authors for their nice reply and edition of manuscript to improve the quality.